Identification of target genes in neuroinflammation and neurodegeneration after traumatic brain injury in rats

Zhao Jianwei
Xu Chen
Cao Heli
Zhang Lin
Wang Xuyang
Chen Shiwen chenshiwen@126.com
Department of Neurosurgery, Shanghai Jiao Tong University Affiliated Sixth People’s Hospital , Shanghai , China
Uversky Vladimir
Electronic publication date: 2019 Dec 19
Publication date: 2019
Volume: 7
Electronic Location ID: e8324
Received 2019 Sep 13; Accepted 2019 Dec 1
Copyright: © 2019 Zhao et al.
Copyright year: 2019
Copyright holder: Zhao et al.
License: This is an open access article distributed under the terms of the Creative Commons Attribution License, which permits unrestricted use, distribution, reproduction and adaptation in any medium and for any purpose provided that it is properly attributed. For attribution, the original author(s), title, publication source (PeerJ) and either DOI or URL of the article must be cited.
License URL: https://creativecommons.org/licenses/by/4.0/

Keywords: Traumatic brain injury, Integrated analysis, Neuroinflammation, Neurodegeneration, Bioinformatics

Funding: Shanghai Science and Technology Commission 19ZR1438600 This work was supported by the project of Shanghai Science and Technology Commission (19ZR1438600). The funders had no role in study design, data collection and analysis, decision to publish, or preparation of the manuscript.

==============================
Background

Traumatic brain injury (TBI) is a common neurological emergency observed in hospitals. A considerable number of patients suffer from long-term disabilities after TBI. This study aimed to identify altered gene expression signatures and mechanisms related to TBI-induced chronic neuroinflammation and neurodegeneration.

Methods

An integrated analysis was performed using published RNA-sequencing studies to determine TBI-induced differentially expressed genes (DEGs). Based on the DEG data, functional annotation, signal-net, and transcription factor analyses were conducted to understand the mechanism of chronic neuroinflammation and neurodegeneration induced after TBI.

Results

Two datasets were obtained using the Gene Expression Omnibus database, of which, 6,513 DEGs were identified (6,464 upregulated and 49 downregulated). Positive regulation of biological process, positive regulation of cellular process, nucleus, and heterocyclic compound binding were Gene Ontology terms significantly enriched in post-TBI rat models. Leukocyte transendothelial migration, chemokine signaling pathway, neurotrophin signaling pathway, and longevity-regulating pathway were significantly enriched after TBI. With regard to the signal-net analysis, FOXO3, DGKZ and ILK were considered the most critical genes derived using high–betweenness centrality calculation. A total of 44 TFs, including FOXO1, SRY and KLF4, were predicted to play an important role in the upregulation of gene expression. Using integrated bioinformatics analysis, TBI was found to be associated with a significant inflammatory response and neurodegeneration. FOXO3, apolipoprotein (APOE), microtubule-associated protein tau (MAPT), and TREM2 were probably associated with the TBI pathological process. The mitochondrial electron transport chain may be associated with neurodegeneration in patients with TBI, serving as a potential therapeutic target.

Introduction

More than 50 million people worldwide suffer from traumatic brain injury (TBI) each year. Astonishingly, about half of the world’s population may encounter one or more brain injuries in their lifetime (Maas et al., 2017). Based on the World Health Organization estimate, TBI will be the third leading cause of death and disability by the year 2020 (The Lancet Neurology, 2012). Patients who survive TBI most likely suffer from disabilities, including cognitive, motor, and emotional deficits. This has a substantial impact on the health care industry, patients’ families, and society (Puntambekar et al., 2018).

Resident microglia and peripheral macrophages induce inflammation of the central nervous system (CNS) after TBI (Russo & McGavern, 2016). Immune-mediated neuroinflammation can persist for several years after a single brain injury (McKee & Lukens, 2016; Ramlackhansingh et al., 2011). Increasing experimental evidence has demonstrated that the neuroinflammatory system can significantly influence clinical prognosis following TBI. Furthermore, TBI-induced inflammation and pathology have been strongly associated with increased risks of neurodegenerative diseases such as chronic traumatic encephalopathy, Alzheimer’s disease (AD), and Parkinson’s disease (PD) (Goldman et al., 2006; McKee et al., 2009; Mortimer et al., 1991). Numerous epidemiological studies on patients with post-TBI dementia have supported the association between TBI and AD (Nordstrom et al., 2014; Plassman et al., 2000). An understanding of the disease mechanism that results in long-term disabilities after TBI is urgently needed.

An integrated transcriptome analysis was performed in this study using samples from chronic stages of TBI to eliminate transcriptomic changes associated with acute TBI post-injury. Using bioinformatics methods, this study aimed to identify key gene expression changes, pathways, and transcription factors (TFs) associated with chronic changes after TBI. In addition, a signal-net map was constructed to determine key gene interactions. Among the differentially expressed genes (DEGs), genes related to neurodegenerative diseases, especially AD, and neuroinflammation-related genes were the major focus. Besides, apoptosis and autophagy have been widely investigated in TBI and neurodegenerative disorder models (Ghavami et al., 2014; Zhang et al., 2016). This study also explored whether apoptosis and autophagy play a role in the chronic process of brain injury. It described selected genes and pathways relevant to neurodegenerative and neuroinflammation conditions and might provide insights into understanding TBI-induced disabilities.

Materials and Methods

Selection of relevant sequencing data for TBI transcriptomics analysis

The Gene Expression Omnibus (GEO) database (http://www.ncbi.nlm.nih.gov/geo) from the National Center of Biotechnology Information (NCBI) was searched for relevant sequencing datasets for TBI. Datasets were selected based on the following criteria: the samples analyzed were from the hippocampus tissue of Sprague–Dawley (SD) rats and RNA sequencing was performed 3 months after TBI. Two independent reviewers extracted the data from the original published studies. The following data were extracted from each study: GEO accession number, platform, number of rats in the TBI and normal control (NC) groups, and gene expression data.

Identification of DEGs

The raw data of selected studies were obtained from the NCBI-SRA database (https://www.ncbi.nlm.nih.gov/sra/). The HISAT2 (http://ccb.jhu.edu/software/hisat2, v2 2.1.0) was used to align the RNA-seq data and StringTie (http://ccb.jhu.edu/software/stringtie, v1.3.5) to assemble and quantify the transcripts (Pertea et al., 2016). R statistical software (R Core Team, 2018) and Bioconductor package helped visualize the results. The datasets were then assigned into two groups: exp (TBI) group and con (control) group. The DEGs between the TBI hippocampus tissue and control tissue were analyzed using the classical t test (stats package, version 3.4.4) in R (settings: P < 0.05; false discovery rate (FDR) < 0.05; fold change (FC) > 1.2; or fold change < 0.833). A heat map for top 50 DEGs was produced using the “pheatmap” package (version 1.0.10) of R. A principal component analysis (PCA) plot was conducted to determine the principal components using the DEG list and visualized using the “ggbiplot2” R package.

Functional and pathway enrichment analysis of the DEGs

Functional terms were retrieved from the Gene Ontology (GO) database. The analyses were performed using the Fisher’s exact test and multiple comparison test, with a P value < 0.05 set as the cutoff criteria. A pathway analysis was performed to identify significantly different regulatory pathways using the Kyoto Encyclopedia of Genes and Genomes (KEGG). Fisher’s exact test and χ2 test were used to select significantly enriched pathways. GO and KEGG analyses were performed using the clusterProfiler package (http://bioconductor.org/biocLite.R, version 3.10.1) of R statistical software.

Signal-net analysis

The KEGG database was used to analyze functional gene interactions (Schlitt et al., 2003) and gene signal transduction network (signal-net) (Zhang et al., 2017) to analyze the connectivity of DEGs and their potential role in TBI. Intergenic connections between upstream and downstream genes included activation, phosphorylation, binding, compound, and dephosphorylation. Within the signal-net map, nodes indicated genes and edges indicated relationship types between the DEGs. The significance of a gene in the network was evaluated by measuring its “between centrality.” “Between centrality” represented the mediating capacity of each gene, that is, the number of times a node was located on the shortest path between two other nodes. Hence, the higher the between centrality score for a gene, the greater its significance in the signal-net map. The Cytoscape (3.7.2) software was used to generate the signal-net analysis map.

Identification of TFs

Further, 2,000 bp upstream and 500 bp downstream from the transcription start site (TSS) of each DEG was analyzed for TF binding sites. Based on the Match algorithm, the TRANSFAC database was used to predict possible TF binding sites adjacent to the TSS. Prediction results were evaluated using core and matrix scores. Core match referred to five conserved bases of the binding region between the TF and the promoter, which were indicated with capital letters. Matching score indicated the accuracy of prediction results. Higher values indicated higher precise prediction results. Matrix scores indicated the accuracy of prediction results, with higher scores indicating higher precise prediction results. The integrated analysis predicted the TFs of selected genes associated with neurodegeneration and inflammation. Besides, the network map of mRNA-TF was constructed using the Cytoscape (3.7.2) software.

Results

Selection of sequencing datasets for TBI transcriptome analysis

The gene expression profiles of GSE75120 (Debski et al., 2016) and GSE80174 (Lipponen et al., 2016) included 10 TBI and 10 NC samples. These two sequencing datasets provided gene expression profiles of the hippocampus tissue of male SD rats. TBI was induced using the lateral fluid percussion (LFP) method. The hippocampal tissue was profiled 3 months after LFP-induced TBI with corresponding sham-treated controls. High-throughput sequencing was performed for the two studies using the Illumina Genome Analyzer IIx (Rattus norvegicus) platform.

Identification of DEGs

A total of 6,513 genes were identified to be differentially expressed between TBI and the NC group; of these, 6,464 were upregulated and 49 were downregulated (P < 0.05, FDR < 0.05 and FC > 1.2 or FC < 0.833). The normalized raw data and DEG data files are available in the Supplemental Files. A PCA plot was drawn to assess the variability of data. The PC1 and PC2 explained 74.6% and 5.2% of the variance in the data, respectively. Exp group was isolated from the con group. For a more in-depth exploration, 20 genes were listed, including six hub genes obtained in the signal-net analysis and 14 genes related to neurodegeneration and neuroinflammation. Further, 20 genes of interest were selected (Table 1). A heat map of all the DEGs is depicted in Fig. 1A, whereas a PCA plot of DEGs is depicted in Fig. 1B.

Table 1 DEGs (selected) between TBI and NC.

Gene symbol	P_values	P_adjust	Fold_change (Exp/Con)	Style	
Apoe	0.000606	0.027409	1.778066	up	
Plcb3	0.001583	0.029642	1.76812	up	
Tyrobp	0.002017	0.029917	2.243941	up	
Dgkz	0.002115	0.030317	1.656656	up	
Mag	0.002141	0.030421	1.608197	up	
Abca1	0.002142	0.030421	1.704966	up	
Ilk	0.002785	0.03094	1.410699	up	
Trem2	0.002807	0.031055	1.830888	up	
Fcgr3a	0.003078	0.031063	1.685427	up	
Foxo3	0.00573	0.033484	1.514945	up	
Abi3	0.005796	0.033605	1.633516	up	
Rxra	0.006317	0.034032	1.710812	up	
Map1lc3a	0.0075	0.035039	1.430401	up	
P2ry12	0.008467	0.036137	1.60919	up	
Sqstm1	0.008739	0.036397	1.504762	up	
Mapt	0.009304	0.037017	1.451565	up	
Tmem119	0.010333	0.038487	1.756732	up	
Slco2b1	0.011369	0.039789	1.614069	up	
Calm1	0.019664	0.048272	1.392877	up	
Casp9	0.01971	0.048307	1.45513	up	

Figure 1 (A) Heat map for top 50 DEGs between TBI versus control; (B) PCA plot for all DEGs between TBI versus control.

Exp: TBI group, marked in red; Con: Control group,marked in blue.

Functional enrichment analysis for DEGs

Three functional groups were classified based on the GO analysis: cellular component (CC), molecular function (MF), and biological process (BP). In the MF group, heterocyclic compound binding, organic cyclic compound binding, ion binding, and RNA binding were the most significantly enriched GO terms (Fig. 2). With regard to CC, the most enriched GO terms were nucleus, cytosol, intracellular organelle lumen, and membrane-enclosed lumen. For BP analysis, DEGs were significantly enriched for the positive regulation of the BP, positive regulation of the cellular process, and CC organization or biogenesis. The top GO term results enriched using DEGs for TBI are shown in Fig. 2. After the KEGG pathway enrichment analysis of the DEGs, 15 significantly enriched molecular pathways of interest were selected. These included leukocyte transendothelial migration, chemokine signaling pathway, neurotrophin signaling pathway, and longevity-regulating pathway (Table 2; Fig. 3). On directly comparing the overlapping gene list from KEGG, genes that overlapped with neurodegenerative brain disorders, including AD, PD, and Huntington disease, were found (overlapping genes shown in Table 3).

Figure 2 Top 10 most significantly enriched GO terms in the three functional groups (compared to the NC).

Figure 3 15 KEGG pathways of interests in TBI.

Y‐axis represents the pathway and the X‐axis represents the negative logarithm of P value (−LgP). The higher the value, the smaller the P value, that is, the higher the significance level of the differential gene pathway. The color and size of the bubble represents −LgP significance and the number of differentially expressed genes enriched in the pathway, respectively.

Table 2 KEGG pathway analysis.

Pathway	P-value	Enrichment score	Overlapping gene count	
Apoptosis	2.36733E−09	1.753575	79	
MAPK signaling pathway	1.62509E−07	1.439707	138	
Metabolic pathways	3.50457E−05	1.14622	501	
Leukocyte transendothelial migration	0.000123951	1.524076	56	
p53 signaling pathway	0.000392348	1.607194	38	
Glioma	0.000752699	1.58694	36	
Neurotrophin signaling pathway	0.00090182	1.427188	57	
PI3K-Akt signaling pathway	0.002774279	1.219635	136	
Ras signaling pathway	0.003476288	1.26244	96	
Mitophagy—animal	0.005672088	1.492673	31	
Autophagy—animal	0.006088794	1.331332	57	
NF-kappa B signaling pathway	0.006825466	1.387437	43	
Longevity regulating pathway	0.012090795	1.371485	39	
TNF signaling pathway	0.013180888	1.32523	47	
Chemokine signaling pathway	0.014932003	1.245003	72	
NOD-like receptor signaling pathway	0.019384236	1.244378	66	

Table 3 Overlapping genes for neurodegenerative diseases.

Pathway	Overlap gene	
Alzheimer disease	Casp9 Cox4i1 Cox4i2 Cox5a Cox5b Cox6a1 Cox6b1 Cox6b2 Cox8a Ndufa1 Ndufa11 Ndufa12 Ndufa4l2 Ndufa6 Ndufa7 Ndufa8 Ndufa9 Ndufb1 Ndufb10 Ndufb2 Ndufb6 Ndufb7 Ndufb8 Ndufb9 Ndufc2 Ndufs2 Ndufs5 Ndufs7 Ndufs8 Ndufv1 Ndufv3 Sdhb Sdhd Uqcrc1 Uqcrq	
Parkinson disease	
Huntington disease	

Signal-net analysis of the DEGs

To further investigate gene interaction networks, the most significant gene nodes were identified using the signal-net analysis based on the genes of interest. FOXO3, DGKZ, ILK, RXRA, PLCB3 and CALM1 were found to be the most significant genes determined using high–betweenness centrality calculation (Fig. 4).

Figure 4 Signal-net map.

Circles represent genes (red indicates hub genes, while blue indicates the other genes). The size of the circle represents the level of “betweenness centrality.” Lines represent interactions between genes.

TF analysis

A total of 20 DEGs (DEGs enriched for pathways related to neurodegeneration and neuroinflammation, shown in Table 1) were used to construct the mRNA-TF-Net (Fig. 5). Based on the map, 44 TFs were predicted to play an important role in the upregulation of gene expression. FOXO1 and SRY (degree = 20), and KLF4, KLF6, PARP1, POU3F3, PRRX1 and RUNX1 (degree = 19) were the top TFs found to play a role in DEGs.

Figure 5 mRNA-TF-Net.

Blue represents mRNAs, red represents top TFs, and yellow represents other TFs.

Discussion

An integrated analysis of high-throughput sequencing data was performed to determine the molecular mechanisms associated with TBI. Using the bioinformatics analysis, several genes, pathways, and TFs associated with neuroinflammation and neurodegeneration, which might play important roles in post-TBI molecular mechanisms, were identified.

The present study was consistent with previous studies demonstrating that TBI induced neuroinflammation and neurodegeneration (Meng et al., 2017; Ransohoff, 2016; Russo & McGavern, 2016; Simon et al., 2017; White et al., 2013). Several genes enriched for TBI and neurodegenerative diseases were identified. Apolipoprotein (APOE), an independent risk factor for the development of AD, was found to be upregulated (Farrer et al., 1997; Lupton et al., 2016). In addition, the microtubule-associated protein tau (MAPT) gene, a neuronal marker, was significantly upregulated in the TBI group. The MAPT gene encodes the tau protein and is involved in multiple neuropathologies, especially AD (Garcia-Escudero et al., 2017).

TREM2 has been demonstrated to be an independent risk factor for late-onset AD (Guerreiro et al., 2013). The R47H variant in TREM2 is associated with a significant increase in AD risk (Guerreiro et al., 2013). In the present analysis, TREM2 and TYROBP (encode DAP12, an adaptor that regulates signaling via TREM2) (Thrash, Torbett & Carson, 2009) were upregulated in hippocampal samples from rats with 3-month post-TBI. Furthermore, FOXO3 was found to be an important gene based on high–betweenness centrality calculation after the signal-net analysis. Several studies have consistently demonstrated that APOE and FOXO3 function as “longevity genes” (Broer et al., 2015; Willcox et al., 2006). Deacetylation of FOXO3A has been shown to possess a neuroprotective role in Huntington’s disease models (Jiang et al., 2011). Besides TREM2, the expression of ABI3 also increased in TBI according to the present analysis, which has been considered to modulate the susceptibility to AD in recent years (Dalmasso et al., 2019). Both of them are highly expressed in microglia, providing evidence for the role of a microglia-mediated innate immune response in the development of AD (Sims et al., 2017).

The activation of microglia residing in the CNS and subsequent recruitment of peripheral inflammatory macrophages play a major role in acute immune response after TBI; it lasts for months to years (McKee & Lukens, 2016). Hickman et al. (2013) demonstrated that both brain-resident microglia and peripheral macrophages express a unique set of genes and are distinct from other cells. Several genes exclusively expressed in microglia were found with significantly higher expression compared with controls. They included SLCO2B1, TMEM119, P2RY12, FCGR3A, TYROBP and TREM2. Additionally, genes exclusively expressed in macrophages were also significantly upregulated after TBI and included PF4, CRIP1 and PRG4. These results indicated that the chronic activation of infiltrating macrophages and resident microglia were present in the hippocampus for at least 3 months after TBI.

Apoptosis and autophagy are the basic physiological processes to maintain brain homeostasis. In the present study, the autophagy markers (SQSTM1 and MAP1LC3A) and the apoptosis marker (CASP9) both increased significantly in TBI. However, the relationship between them is perplexing due to the interconnected nature of both processes (Zhang et al., 2013). Hence, the regulation of apoptosis and autophagy processes should be done with caution. Also, further studies are needed to clarify the role of both processes in the chronic course of TBI. Axonal repair is important for the recovery of brain injury. ABCA1 is a major regulator of cellular cholesterol and phospholipid homeostasis necessary for axonal restoration, which transports lipids and cholesterol onto APOE (Castranio et al., 2018). Meanwhile, MAG, which is a transmembrane glycoprotein localized in periaxonal cells, plays a role in the inhibition of nerve regeneration after injury in the CNS (Yiu & He, 2006). In the present study, the expression of ABCA1 and MAG both increased significantly in chronic brain injury. Related target gene therapy may be effective for axonal repair in TBI.

Then, KEGG enrichment analysis was performed, focusing on pathways associated with neurodegeneration and neuroinflammation. Neurodegenerative diseases, such as PD, AD, and Huntington’s disease, were not significantly enriched. However, several overlapping genes were identified for these three neurodegenerative diseases. These included NADH dehydrogenase (ubiquinone) subunits such as Ndufa1, Ndufa11, Ndufa12 and Ndufa6, and cytochrome c oxidase subunits such as Cox4i1, Cox5a and Cox6a1. Large protein complexes are found in the electron transport chain (ETC) located in the mitochondria. TBI may induce the expression of genes in the mitochondrial ETC in various brain regions (Xing et al., 2013). ETC overexpression in the hippocampus has been reported to be associated with aging and increased oxidative damage in mouse brains (Manczak et al., 2005), and with age-related diseases such as AD and PD (Kim et al., 2000; Manczak et al., 2005). The data suggested that increased ETC expression in mitochondria might be associated with poor prognoses, such as neurodegenerative disorders in patients with TBI. Studies on patients with TBI and animal models have demonstrated an imbalance in immune response, which subsequently leads to neurological dysfunction and brain pathology disorders (Shiozaki et al., 2005; Woiciechowsky et al., 2002). Immune-mediated inflammatory responses can last for months to several years following initial injury (Jassam et al., 2017; Johnson et al., 2013) and result in chronic tissue damage (Puntambekar et al., 2018). Leukocyte transendothelial migration, NOD-like receptor signaling pathway, and chemokine signaling pathway related to the immune system were enriched after TBI. In addition, pathways that might be neuroprotective, such as neurotrophin signaling pathway, autophagy pathway, and longevity-regulating pathway, were also upregulated.

TBI-induced behavioral changes are associated with duration after primary trauma (Baratz et al., 2011). The data of the present study supported this observation because rats with TBI displayed significant changes in gene sets associated with neuroinflammation and neurodegenerative disease 3 months after TBI. TBI pathology has been demonstrated to be associated with inflammation; however, currently, no consensus exists regarding whether and how it should be targeted therapeutically. The relationship between neuroinflammation and neurodegenerative disease is difficult to define. However, understanding the changes that occur in immunocytes, such as in microglia, after TBI might provide clues on how to prevent the chronic pathological process of TBI, including chronic neuroinflammation. Strategies targeting the microglia are believed to be effective in inhibiting neurodegeneration. For example, an increase in the ability of microglia to recognize amyloid-beta peptide (Aβ) and abnormal aggregation of tau filaments (the cardinal features of AD) have been achieved via immunotherapy using specific antibodies (Citron, 2010; Kondo et al., 2015).

One limitation of the present study was that the sequencing data was retrieved using whole hippocampus tissues rather than purified cells. Gene expression changes observed in these studies reflected a mixture of gene expression profiles (glial cells, neurons, and even possibly circulating blood leukocytes). Hence, new techniques such as laser capture microdissection to isolate and capture diverse cell types, are necessary to understand TBI-induced neurodegenerative mechanisms.

Conclusions

In conclusion, the present study demonstrated that TBI was associated with significant inflammatory responses and neurodegeneration. It showed that FOXO3 played a core role in modulating post-BI gene networks. Several DEGs, including APOE, MAPT and TREM2, might be involved in neurodegeneration after TBI, with the participation of microglia and peripheral macrophages in chronic inflammation after TBI. In addition, the data suggested that ETC expression in mitochondria might be associated with neurodegeneration in patients with TBI. The findings of the present study might provide novel insights into mechanisms related to the chronic phase of TBI and may help develop new diagnostic and therapeutic treatments for TBI.

Supplemental Information

Supplemental Information 1 Raw data, codes, and step-by-step data analysis procedures.

Click here for additional data file.

Additional Information and Declarations

Competing Interests

Author Contributions

Data Availability

The authors declare that they have no competing interests.

Jianwei Zhao conceived and designed the experiments, performed the experiments, authored or reviewed drafts of the paper, approved the final draft.

Chen Xu performed the experiments, contributed reagents/materials/analysis tools, prepared figures and/or tables, approved the final draft.

Heli Cao performed the experiments, contributed reagents/materials/analysis tools, authored or reviewed drafts of the paper, approved the final draft.

Lin Zhang analyzed the data, prepared figures and/or tables, approved the final draft.

Xuyang Wang analyzed the data, prepared figures and/or tables, approved the final draft.

Shiwen Chen conceived and designed the experiments, authored or reviewed drafts of the paper, approved the final draft.

The following information was supplied regarding data availability:

The raw data and code files are available in the Supplemental Files.

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
