# Peer review of "Identification of target genes in neuroinflammation and neurodegeneration after traumatic brain injury in rats"

_PeerJ, doi:10.7717/peerj.8324_

## Round 0.1 · original submission · Major Revisions

Please carefully address all the critiques of three reviewers and revised your manuscript accordingly

Reviewer 1 ·

Basic reporting

The manuscript is clearly written, it provides enough background and has the necessary citations.

Experimental design

no comment

Validity of the findings

no comment

Additional comments

In this manuscript Zhao et al identified genes that had altered expression after traumatic brain injury. Using multiple bioinformatics methods, they identified several genes that may play key role in neurodegeneration after brain injury. The manuscript is nicely written and there are only a few minor mistakes that need to be corrected:
1. Line 44: I assume that the (2012) is a reference. Please make sure to cite references correctly.
2. Line 132: The authors state that they selected 20 genes of interest out of 6513 genes. Why these 20 genes? What was the criteria for selecting those 20 genes only? Please provide more details in this section.

Reviewer 2 ·

Basic reporting

In this paper, the researchers have conducted an integrated analysis using published RNA-sequencing studies to determine differential expression of genes induced by traumatic brain injury (TBI), and identified thousands of genes which were relevant. Overall, although the researchers have provided lots of statistic analysis, the novelty of this paper is low and the results were too broad to provide valuable information which needs to be really focused on.

Experimental design

1. Most of the figures did not provide detailed legends to fully describe the figure
2. In Figure 4, it’s hard to distill any useful information since there’s no information about the specific genes. It’s better to revise this figure with more details about the genes, otherwise it’s redundant.
3. In Figure 5, it’s also hard to get really useful information since there’s too much. It’s better to just present a few of most relevant genes

Validity of the findings

N/A

Additional comments

N/A

Reviewer 3 ·

Basic reporting

1, The authors analyzed the gene expression changes using previously published data and concluded that FOXP3, APOE, MAPT and TREM2 are associated with neuroinflammatory and neurodegenerative disorders after TBI based on their analysis.

2, For Fig. 1, the authors should clearly label the data source and treatment in the x-axis, rather than just showing the file names. Also, I think the authors should show the Top50 or Top100 DEG (as well as the gene names) rather than showing all the genes in the heatmap.

3, The reference need to be double-checked. For example, author names for the first reference is missing.

4, For Fig. 4, the authors should at least clearly labeled the gene names of the largest circles so the readers can understand the conclusion made in Page 13, line 153.

Experimental design

1, DEG analysis is critical in this study. Step-by-step procedures regarding how related data analyzed was performed should be clearly written in the methods section. Else, the authors could included their original codes in the supplemental data.

Validity of the findings

1, Raw data of all gene expression should be provided in the supplemental for the readers.

2, As a standard part of DEG analysis, a PCA plot showing a separation of TBI group and Control group will make this study more convincing.

3, The authors selected 20 genes of interest in table 1. Why these 20 genes are selected? Are these genes with highest fold change or p-value?

4, Based on the data provided in this study, I don’t think it’s appropriate or necessary to include the four genes in the title. The authors claimed that they found 6464 up-regulated genes in this study and the fold change of these four genes are less than 2 fold based on table 1.

---

## Round 0.2 · accepted · Accept

Since all critiques were adequately addressed and the manuscript was amended accordingly, I am please to accepted revision.

Reviewer 1 ·

Basic reporting

no comment

Experimental design

no comment

Validity of the findings

no comment

Additional comments

The authors have addressed all my concerns.

Reviewer 3 ·

Basic reporting

The authors well addressed my concerns.

Experimental design

The work is well designed.

Validity of the findings

The findings are interesting and all raw data and methods are provided.